# Quantized convolutional neural networks through the lens of partial differential equations

**Ido Ben-Yair**
Department of Computer Science
Ben-Gurion University of the Negev
Beer-Sheva, Israel
idobeny@post.bgu.ac.il

**Moshe Eliasof**
Department of Computer Science
Ben-Gurion University of the Negev
Beer-Sheva, Israel
eliasof@post.bgu.ac.il

**Eran Treister**
Department of Computer Science
Ben-Gurion University of the Negev
Beer-Sheva, Israel
erant@cs.bgu.ac.il

## Abstract

Quantization of Convolutional Neural Networks (CNNs) is a common approach to ease the computational burden involved in the deployment of CNNs. However, fixed-point arithmetic is not natural to the type of computations involved in neural networks. In our work, we consider symmetric and stable variants of common CNNs for image classification, and Graph Convolutional Networks (GCNs) for graph node-classification. We demonstrate through several experiments that the property of forward stability preserves the action of a network under different quantization rates, allowing stable quantized networks to behave similarly to their non-quantized counterparts while using fewer parameters. We also find that at times, stability aids in improving accuracy. These properties are of particular interest for sensitive, resource-constrained or real-time applications.

## 1 Introduction

Deep neural networks have demonstrated superiority in solving many real world problems. In particular, CNNs are an effective approach for processing structured high-dimensional data. Consequently, the demand for the deployment of CNNs on resource-constrained devices is constantly increasing, where at the same time, CNNs grow larger than ever. CNNs still face critical challenges despite this success and their predictions can be highly sensitive to perturbations of the input [32, 18]. Because they often require extremely deep and wide architectures they often impose a high computational cost and make deployment on resource-constrained devices prohibitive, especially for mission-critical applications such as autonomous driving.

Several authors have established a direct link between CNNs and partial differential equations (PDEs) [12, 8, 21]. The connection is two-fold. First, CNNs filter input features with multiple layers, employing both elementwise non-linearities and affine transformations, which can be seen as linear combinations of the finite difference discretizations of spatial derivative operators [35, 15, 14]. Furthermore, the popular residual network architectures (ResNets) [23] can be interpreted as a discrete time integration of a non-linear ODE using the forward Euler scheme. Another possibility is to view the network as a continuous function [10, 35, 20, 17]. This conceptual framework gives rise to many interesting questions such as: what is the importance of the forward stability of CNNs, and can we

create analogues to stable time integration through Courant-Friedrichs-Lewy conditions, to make CNNs more robust to errors? Specifically, in this work we consider round-off and quantization errors.

Quantization methods enable neural networks to carry out computations with fixed-point operations rather than floating-point arithmetic. This contributes to the efficiency of the networks and reduces their memory footprint, often at the cost of accuracy [11]. Accordingly we consider a per-layer, uniform and static (same number of bits for every layer) quantization-aware training method.

In our work we wish to examine the concept of quantized CNNs by considering that forward time integration in PDEs inevitably generates error at every step. To satisfy forward stability, this error must decay from time step to time step, such that the integration error is bounded, resulting in a discrete approximation close to the real continuous solution of the PDE. Therefore, a significant contribution of our work is the promotion of stability in quantized CNNs via small changes to common architectures. We examine the behaviour of quantization under symmetric and stable, heat equation-like CNNs [21, 1, 2]. We show that the quantization process produces significantly lighter-weight networks, with minimal loss of accuracy, and study the *consistency* of quantization methods in CNNs, a key attribute in the construction of quantized variants of CNNs. To this end, we measure the aforementioned similarity using symmetric (potentially stable) and non-symmetric (potentially unstable) networks. Our experiments indicate that symmetric and stable networks achieve better consistency and that *quantized networks are expected to perform better and more in line with their full-precision counterparts when stable architectures are used, rather than unstable architectures*.

Neural networks are known to be susceptible to noise in their inputs [19]. Similarly, the quantization of the activation maps inevitably adds noise to the input of *every* layer. While the quantization error is not chosen specifically for a given input, it does appear in every layer. The introduction of such errors throughout the layers of the network resembles the approximation and round-off errors in time integration of PDEs which require forward stability to converge. To relieve this phenomenon, we follow the approach of constructing stable architectures that are more robust against such errors [21].

We call a discrete forward propagation *stable* if it prevents any perturbation from growing as it propagates through time steps. More explicitly, an $N$-layered network is stable if there exists some $M > 0$ such that

$$\|\mathbf{x}_N - \tilde{\mathbf{x}}_N\| \leq M \|\mathbf{x}_i - \tilde{\mathbf{x}}_i\|, \quad i = 1, ..., N - 1 \tag{1}$$

where $\mathbf{x}_i$ and $\tilde{\mathbf{x}}_i$ are the true and perturbed feature maps of the $i$-th layer, respectively. The Jacobian of $\mathbf{x}_N$ with respect to $\mathbf{x}_i$ can then be bounded proportionally to $M$, and accordingly, any stable architecture needs to have a bounded Jacobian [35]. See Appendix A for a more precise formulation.

## 2 Stable and quantized residual networks

To promote stability in CNNs and to prevent the amplification of quantization errors, we would like the quantized network to behave similarly (in terms of its activations) to a non-quantized instance of the same network. The symmetric variant of ResNet, together with the activation quantization operator is given as follows:

$$\mathbf{x}_{j+1} = Q_b(\mathbf{x}_j - h\mathbf{K}_j^\top Q_b(\sigma(\mathbf{K}_j\mathbf{x}_j))), \tag{2}$$

where $Q_b$ denotes the quantization operation. We consider the same weights $\mathbf{K}$ between the two architectures, whether quantized or not, and also assume that the absolute quantization error for any scalar is bounded by some $\delta$. We analyze the propagation of the quantization error and show that it is the Jacobian matrix of the ResNet block that multiplies the error at every iteration and propagates the previous error into the next block. Assuming that $\sigma$ is non-decreasing, it means that $\mathbf{K}_j^\top \sigma'()\mathbf{K}_j$ is positive semi-definite, and with a proper choice of $\mathbf{K}_j$ and $h$, we can force $\rho(\mathbf{J}_j) < 1$, so that the error decays. To ensure this forward stability we must set $h < 2(L\|K_j\|_2^2)^{-1}$ for every layer $j$ in the network, where $L$ is the upper bound for $\sigma'()$ [1]. This is generally possible in ResNets only if we use the symmetric variant in of ResNet. We measure the quantization error by

$$\text{MSE}(\mathbf{x}, \mathbf{x}_b) = \frac{1}{n}\sum_{i=1}^{n}(x_i - (x_b)_i)^2, \tag{3}$$

and reduce this MSE to maximize the accuracy of the quantized network, as shown in [4].

Table 1: CIFAR-10 image classification on quantized stable and unstable networks, and their consistency with their non-quantized counterparts.

| Architecture | Params (M) | Accuracy (%) | | | MSE (Acc.) |
|---|---|---|---|---|---|
| | | FP/FP | 4W/8A | 4W/4A | 4A → 32A |
| ResNet56 (orig.) | 0.85 | 93.8 | 93.0 | 92.6 | 0.076 (92.9) |
| Stable ResNet56 (ours) | 0.41 | 92.1 | 92.3 | 91.6 | 0.024 (92.0) |
| MobileNetV2 (orig.) | 2.20 | 94.0 | 92.9 | 92.4 | 0.067 (92.5) |
| Stable MobileNetV2 (ours) | 1.70 | 93.1 | 92.2 | 91.6 | 0.034 (91.7) |

## 3 Stable and quantized GCNs

We also study the importance of stability for quantized Graph Convolution Networks (GCNs), which can be thought of as a PDE discretized on unstructured grids [14]. Specifically, we use the diffusive PDE-GCN architecture formulated in [13], which utilizes symmetric operators, only on unstructured graphs:

$$\mathbf{x}_{j+1} = \mathbf{x}_j - h\mathbf{S}_j^\top \mathbf{K}_j^\top \sigma(\mathbf{K}_j \mathbf{S}_j \mathbf{x}_j). \tag{4}$$

Here, $\mathbf{x}_j$ are the features defined over the nodes of a graph, $\mathbf{K}_j$ is a learnt $1 \times 1$ convolution operator and $\mathbf{S}_j$ is either learnt (e.g., as in [14]) or pre-defined spatial operation (e.g., the graph Laplacian), both for the $j$-th layer of the network. Here, the forward stability is guaranteed in the continuous case by the symmetry of the operator [13]. Refer to Appendix A for a more precise formulation.

We refer to a network governed by the dynamics in Eq. (4) as PDE-GCN$_\text{D}$(sym.). Analogously, we define the non-symmetric residual layer:

$$\mathbf{x}_{j+1} = \mathbf{x}_j - h\mathbf{S}_j^\top \mathbf{K}_{j_2} \sigma(\mathbf{K}_{j_1} \mathbf{S}_j \mathbf{x}_j) \tag{5}$$

Where $\mathbf{K}_{j_1}$ and $\mathbf{K}_{j_2}$ are distinct learnt $1 \times 1$ convolution operators. This formulation does not necessarily yield a symmetric operator, thus, we denote by such a network by PDE-GCN$_\text{D}$(non-sym.). The quantization for Eq. (4) and (5) is applied to the weights and before each convolution operator.

## 4 Image classification using stable and quantized networks

We evaluate our symmetric architectures under quantization by testing them using ResNet-34, ResNet-56 [23] and MobileNetV2 [36]. We conduct several experiments using CIFAR-10/100 and present their results in Tab. 1 and 2. Symmetric networks achieve similar accuracy as non-symmetric networks, while using approximately half of the parameters, a significant saving obtained as a by-product of promoting stability.

Furthermore, we quantify the notion of stability by comparing the behaviour of symmetric and non-symmetric networks under 4-bit quantization. In addition to the 4-bit quantized network, we relax the quantization of the activation maps, allowing them to use the full 32-bit precision, and we measure the divergence between corresponding activations in both runs of the same network - one at 4 bits and the other at 32 bits. This divergence is summarized as the per-entry MSE between the activation maps throughout the two networks. Tables 1 and 2 show that the divergence of the two runs is greater in the non-symmetric (original) network variants, as predicted by the theoretical analysis. This is illustrated in Fig. 1, where we plot the same MSE difference along the layers of each network.

## 5 Semi-supervised node-classification with quantized GCNs

We employ PDE-GCN$_\text{D}$(sym.) and PDE-GCN$_\text{D}$(non-sym.) from Eq. (4)-(5) for semi-supervised node-classification on the Cora, CiteSeer and PubMed data-sets. Namely, we measure the impact of symmetry on the network's accuracy, as well as its ability to maintain similar behaviour with respect to activation similarity under activation quantization of 8 and 4 bits. In all experiments, we use 32 layers, with 64 hidden channels for Cora, and 256 hidden channels for CiteSeer and PubMed. The training and evaluation procedure we used is identical to [28]. Details on initialization and

Table 2: CIFAR-100 image classification on quantized stable and unstable networks, and their consistency with their non-quantized counterparts.

| Architecture | Params (M) | Accuracy (%) | | | MSE (Acc.) |
|---|---|---|---|---|---|
| | | FP/FP | 4W/8A | 4W/4A | 4A → 32A |
| ResNet34 (orig.) | 21.3 | 78.5 | 75.1 | 74.7 | 0.019 (74.7) |
| Stable ResNet34 (ours) | 9.60 | 76.8 | 75.4 | 75.0 | 0.0083 (75.2) |
| ResNet56 (orig.) | 0.86 | 72.0 | 69.0 | 69.6 | 0.14 (70.0) |
| Stable ResNet56 (ours) | 0.41 | 68.4 | 66.7 | 66.1 | 0.031 (67.3) |
| MobileNetV2 (orig.) | 2.30 | 74.2 | 71.1 | 69.8 | 0.083 (70.6) |
| Stable MobileNetV2 (ours) | 1.80 | 73.1 | 71.6 | 70.9 | 0.058 (71.0) |

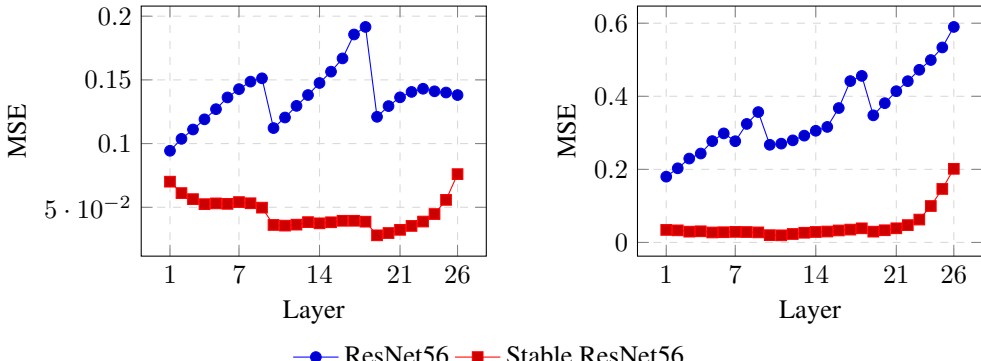

ResNet56 —■— Stable ResNet56

Figure 1: Per-layer MSE between the activation maps of symmetric and non-symmetric network pairs - in each pair one network has quantized activation maps and the other does not. The symmetric variants (red) exhibit a bounded divergence, while the non-symmetric networks diverge as the information propagates through the layers (blue), and hence are unstable. Both networks in each pair achieve similar classification accuracy. ResNet56 on CIFAR-10 (left) and CIFAR-100 (right).

hyper-parameter tuning, are identical to [13]. The results provided in Tab. 3 reveal two benefits of a symmetric (stable) formulation over a non-symmetric (unstable) one. First, it is apparent that that the former results in better accuracy, often by over 2%, while using almost half the number of parameters. In addition, the action of the network is better preserved under quantization using the symmetric formulation. We note that the symmetric formulation in Eq. (4) is a sub-set of the non-symmetric counterpart in Eq. (5). Therefore, theoretically, both networks can achieve identical expressiveness. However, as demonstrated in Tab. 3, this was not observed in our experiments. We attribute this gap to the smoother optimization process of the stable PDE-GCN$_D$(sym.).

Table 3: PDE-GCN$_D$ symmetric vs non-symmetric. All networks are of 32 layers and and float-precision weights. Activation quantization of 8 and 4 bits is applied.

| Data-set | Architecture | Params (M) | Accuracy (%) | | | MSE |
|---|---|---|---|---|---|---|
| | | | FP | 4W/8A | 4W/4A | 4A → 32A |
| Cora | PDE-GCN$_D$ (non-sym.) | 0.35 | 82.7 | 82.2 | 75.7 | 6.11 |
| | PDE-GCN$_D$ (sym.) | 0.22 | 84.3 | 84.0 | 79.4 | 2.03 |
| CiteSeer | PDE-GCN$_D$ (non-sym.) | 5.14 | 73.9 | 72.6 | 71.1 | 20.48 |
| | PDE-GCN$_D$ (sym.) | 3.04 | 75.6 | 74.1 | 72.2 | 12.44 |
| PubMed | PDE-GCN$_D$ (non-sym.) | 4.32 | 79.0 | 79.3 | 75.1 | 5.14 |
| | PDE-GCN$_D$ (sym.) | 2.22 | 80.2 | 80.1 | 77.6 | 2.52 |

## 6 Summary

In this work, we explored quantized neural networks from the perspective of PDEs and demonstrated that stability preserves the action of a network under different quantization rates. We find that at times, stability aids to improve accuracy. These properties are of particular interest for resource-constrained, low-power or real-time applications like autonomous driving.

## Acknowledgments

The research reported in this paper was supported by the Israel Innovation Authority through Avatar consortium, and by grant no. 2018209 from the United States - Israel Binational Science Foundation (BSF), Jerusalem, Israel. ME is supported by Kreitman High-tech scholarship.

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

# A Continuous neural networks and stable ResNet architectures

The general goal of supervised machine learning is to model a function $f : \mathbb{R}^n \times \mathbb{R}^p \to \mathbb{R}^m$ and train its parameters $\boldsymbol{\theta} \in \mathbb{R}^p$ such that

$$f(\mathbf{y}, \boldsymbol{\theta}) \approx \mathbf{c} \tag{6}$$

for input-output pairs $\{(\mathbf{y}^i, \mathbf{c}^i)\}_{i=1}^s$ from a certain data set $\mathcal{Y} \times \mathcal{C}$. Typical tasks include regression or classification, and the function $f$ can be viewed as an interpolation of the true function that we want the machine to learn.

In learning tasks involving images and videos, the learnt function $f$ commonly includes a CNN that filters the input data. Among the many architectures we focus on feed-forward networks of a common type called by ResNets [23]. In the simplest case, the forward propagation of a given example $\mathbf{y}$ through an $N$-layer ResNet can be written as

$$\mathbf{x}_{j+1} = \mathbf{x}_j + hF(\mathbf{x}_j, \boldsymbol{\theta}_j), \quad j = 0, \dots, N - 1, \quad \mathbf{x}_0 = F_{\text{open}}(\mathbf{y}, \boldsymbol{\theta}_{\text{open}}). \tag{7}$$

The layer function $F$ consists spatial convolution operators parameterized by the weights $\boldsymbol{\theta}_1, \dots, \boldsymbol{\theta}_N$, and non-linear element-wise activation functions. The parameter $h > 0$ in this notation serves as a time step, arising from a discretization of a continuous network, as we show in the next section. The classical ResNet layer reads

$$F_{\text{ResNet}}(\mathbf{x}, \boldsymbol{\theta}) = \mathbf{K}_1 \sigma(\mathbf{K}_2 \mathbf{x}). \tag{8}$$

where $\mathbf{K}_1, \mathbf{K}_2$ are two different convolution operators parameterized by $\theta$, and $\sigma$ is a non-linear activation function, like $\sigma(x) = \max\{x, 0\}$, known as the Rectified Linear Unit (ReLU) function. The term $F_{\text{open}}$ denotes an opening layer that typically outputs a $n_c$-channel image $\mathbf{x}_0$, where $n_c$ is greater than the number of input channels, which is typically 3 for RGB images, while possibly also reducing the size of the image. Usually, the opening layer reads

$$\mathbf{x}_0 = \sigma(\mathbf{K}_{\text{open}} \mathbf{y}) \tag{9}$$

where $\mathbf{K}_{\text{open}}$ is a convolution operator parameterized by $\boldsymbol{\theta}_{\text{open}}$ that widens the number of channels from this point onward in the network.

Continuous neural networks have recently been suggested as an abstract generalization of the more common discrete network, where the network is viewed as a discretized instance of a continuous ODE or PDE. As shown by [8, 12, 21, 10], Eq. (7) (or a ResNet) is essentially a forward Euler discretization of a continuous non-linear ODE

$$\partial_t \mathbf{x}(t) = F(\mathbf{x}(t), \boldsymbol{\theta}(t)), \quad t \in [0, T], \quad \mathbf{x}(0) = \mathbf{x}_0 = F_{\text{open}}(\mathbf{y}, \boldsymbol{\theta}_{\text{open}}), \tag{10}$$

where $[0, T]$ is an artificial time interval related to the depth of the network. Relating spatial convolution filters with differential operators, the authors of [35] propose a layer function representation $F$ that renders Eq. (10) similar to a parabolic diffusion-like PDE

$$F_{\text{sym}}(\mathbf{x}, \boldsymbol{\theta}) = -\mathbf{K}^\top \sigma(\mathbf{K} \mathbf{x}). \tag{11}$$

For example, when $\mathbf{K}$ represents a discrete gradient operator and $\sigma(x) = x$, we obtain the heat equation under this treatment. The approach of Eq. (11) is natural, as similar developments have led to several breakthroughs in image processing, including optical flow models for motion estimation [24], non-linear anisotropic diffusion models for image denoising [33, 34], and variational methods for image segmentation [3, 7]. To best balance the network's representational abilities with its computational cost, a "bottleneck" structure is used commonly, exploiting a different number of input and output channels in $\mathbf{K}$, as in Eq. (11).

Generally speaking, the training of the neural network model consists of finding parameters $\boldsymbol{\theta}$ such that Eq. (6) is approximately satisfied for examples from a training data set. The same should also hold for examples from a validation data set, which is not used to optimize the parameters. The training objective is commonly modeled as an expected loss to be minimized, denoted as

$$\min_{\boldsymbol{\theta}} \Phi(\boldsymbol{\theta}), \quad \text{where} \quad \Phi(\boldsymbol{\theta}) = \frac{1}{s} \sum_{k=1}^s \text{loss}(f(\mathbf{y}^k, \boldsymbol{\theta}), \mathbf{c}^k) + R(\boldsymbol{\theta}). \tag{12}$$

Here, $(\mathbf{y}^1, \mathbf{c}^1), \dots, (\mathbf{y}^s, \mathbf{c}^s) \in \mathcal{Y} \times \mathcal{C}$ are the training data, $f(\mathbf{y}, \boldsymbol{\theta})$ includes the action of the neural network, but may also contain other layers, e.g., fully-connected layers, opening and connective layers, and softmax transformations in classification. $R$ is a regularization term. The optimization problem in Eq. (12) is typically solved with gradient-based non-linear optimization techniques [6]. Due to the large-scale and stochastic nature of the learning problem, it is common to use stochastic approximation schemes such as variants of stochastic gradient descent (SGD) like Adam [27]. These methods perform iterations using gradient information from randomly chosen subsets of the data.

## B  Stable and quantized residual networks

As discussed earlier, our goal is to promote stability in CNNs, to prevent the amplification of quantization errors. To obtain that, we would like the quantized network to behave similarly (in terms of its activations) to a non-quantized instance of the same network. The symmetric variant of ResNet in Eq. (8) and (11), together with the activation quantization operator in Eq. (18) is given as follows:

$$\mathbf{x}_{j+1} = Q_b(\mathbf{x}_j - h\mathbf{K}_j^\top \, Q_b(\sigma(\mathbf{K}_j\mathbf{x}_j))), \tag{13}$$

where $Q_b$ denotes the quantization operation in Eq. (18). Let us denote the feature maps of the non-quantized network by $\hat{\mathbf{x}}_j$ (i.e., assuming $q_b(x) = x$ in Eq. (18)), and let the error between corresponding activations be denoted by $\eta_j = \mathbf{x}_j - \hat{\mathbf{x}}_j$. We consider the same weights $\mathbf{K}$ between the two architectures, whether quantized or not, and also assume that the absolute quantization error for any scalar is bounded by some $\delta$. We analyze the propagation of the quantization error $\eta_{j+1}$ as a function of $\eta_j$.

We start unwrapping the block in Eq. (2). First, we subtract $\hat{\mathbf{x}}_{j+1}$ from both sides, and replace the outer quantization with the error term $\eta_{j_1}$.

$$\eta_{j+1} = \eta_{j_1} + \mathbf{x}_j - h\mathbf{K}_j^\top Q_b(\sigma(\mathbf{K}_j\mathbf{x}_j)) - \hat{\mathbf{x}}_{j+1} \tag{14}$$

Next, we remove the other instance of $Q_b$ and add another error term $\eta_{j_2}$:

$$\eta_{j+1} = \eta_{j_1} - h\mathbf{K}_j^\top \eta_{j_2} + \mathbf{x}_j - h\mathbf{K}_j^\top \sigma(\mathbf{K}_j\mathbf{x}_j) - \hat{\mathbf{x}}_{j+1}. \tag{15}$$

Remembering that $\mathbf{x}_j = \hat{\mathbf{x}}_j + \eta_j$, and using the first-order approximation of $\sigma$, we note that $\sigma(\mathbf{K}_j\mathbf{x}_j) \approx \sigma(\mathbf{K}_j\hat{\mathbf{x}}_j) + \sigma'(\mathbf{K}_j\hat{\mathbf{x}}_j)\mathbf{K}_j\eta_j$. Therefore, we have

$$
\begin{aligned}
\eta_{j+1} &= \eta_{j_1} - h\mathbf{K}_j^\top \eta_{j_2} + \hat{\mathbf{x}}_j + \eta_j - h\mathbf{K}_j^\top(\sigma(\mathbf{K}_j\hat{\mathbf{x}}_j) + \sigma'(\mathbf{K}_j\hat{\mathbf{x}}_j)\mathbf{K}_j\eta_j) - \hat{\mathbf{x}}_{j+1} \\
&= \eta_{j_1} - h\mathbf{K}_j^\top \eta_{j_2} + \eta_j - h\mathbf{K}_j^\top \sigma'(\mathbf{K}_j\hat{\mathbf{x}}_j)\mathbf{K}_j\eta_j \\
&= (\mathbf{I} - h\mathbf{K}_j^\top \sigma'(\mathbf{K}_j\hat{\mathbf{x}}_j)\mathbf{K}_j)\eta_j + \eta_{j_1} - h\mathbf{K}_j^\top \eta_{j_2}.
\end{aligned} \tag{16}
$$

The key ingredient of the analysis above is that $\eta_{j_1}$ and $\eta_{j_2}$ are fixed and bounded for every layer. On the other hand, it is the Jacobian matrix of the block $\mathbf{J}_j = \mathbf{I} - h\mathbf{K}_j^\top \sigma'(\mathbf{K}_j\hat{\mathbf{x}}_j)\mathbf{K}_j$ that multiplies $\eta_j$ at every iteration and propagates the previous error into the next block. Assuming that $\sigma$ is non-decreasing, it means that $\mathbf{K}_j^\top \sigma'()\mathbf{K}_j$ is positive semi-definite, and with a proper choice of $\mathbf{K}_j$ and $h$, we can force $\rho(\mathbf{J}_j) < 1$, so that the error decays. To ensure this forward stability we must set $h < 2(L\|K_j\|_2^2)^{-1}$ for every layer $j$ in the network, where $L$ is the upper bound for $\sigma'()$ [1]. This is generally possible in ResNets only if we use the symmetric variant in Eq. (11). [35, 40] also achieved a similar stability result, but not in the context of quantization.

## C  Quantization-aware training

As mentioned before, in this paper we focus on quantized neural networks, and in particular, on the scenario of quantization-aware training. We restrict the values of the weights to a smaller set, so that after training, the calculation of a prediction by the network can be carried out in fixed-point integer arithmetic. The intermediate hidden layers are quantized (rounded) as well. Even though the quantization involves a discontinuous rounding function, the discrete weights can be optimized using gradient-based methods, also known as quantization-aware training schemes [22, 39]. During the forward pass, both the weights and hidden activation maps are quantized, while during gradient calculation in the backward pass, derivative information is passed through the rounding function, whose exact derivative is zero. This method is known as the Straight Through Estimator (STE) [5]. We note that the gradient-based optimization for the weights is applied in floating-point arithmetic as we describe next, but during inference, the weights and activations are quantized, and all operations are performed using integers only.

We now present the details of the quantization scheme that we use, based on [30]. First, we define the pointwise quantization operator:

$$q_b(t) = \frac{\text{round}((2^b - 1) \cdot t)}{2^b - 1}, \tag{17}$$

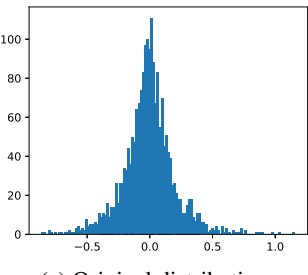
(a) Original distribution

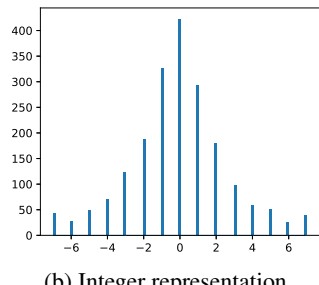
(b) Integer representation

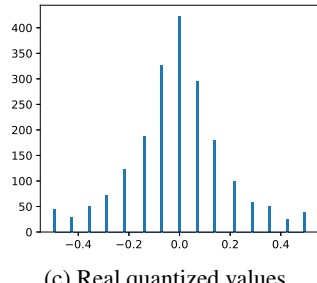
(c) Real quantized values

Figure 2: An example of a uniform signed 4-bit quantized weight tensor. The original distribution is given in (a). The original values are then clipped to the range $[-\alpha, \alpha]$ (in this example $\alpha = 0.5$), divided by $\alpha$ and multiplied by 7 ($= 2^{4-1} - 1$, for 4 bits). Then, the values are quantized to integer values in (b), and scaled back (multiplied by $\alpha/7$) to their original values in (c).

where $t$ is a real-valued scalar in one of the ranges [-1, 1] or [0, 1] for signed or unsigned quantization[1], respectively. $b$ is the number of bits that are used to represent $t$ as integer at inference time (during training, $t$ and $q_b(t)$ are real-valued). During the forward pass, we force each quantized value to one of the ranges above by applying a clipping function to the quantized weights and activations before applying Eq. (17):

$$
\begin{aligned}
w_b &= Q_b(w) = \alpha_w q_{b-1}(\mathrm{clip}(\frac{w}{\alpha_w}, -1, 1)) \\
x_b &= Q_b(x) = \alpha_x q_b(\mathrm{clip}(\frac{x}{\alpha_x}, 0, 1)).
\end{aligned}
\tag{18}
$$

Here, $w, w_b$ are the real-valued and quantized weight tensors, $x, x_b$ are the real-valued and quantized input tensors, and $\alpha_w, \alpha_x$ are their associated clipping parameters (also called scales), respectively. An example of Eq. (18) applied to a weight tensor using 4-bit signed quantization, is given in Fig. 2. During training, we iterate on the floating-point values of the weights $w$, while both the weights and activation maps are quantized in the forward pass (i.e., $x_b$ and $w_b$ are passed through the network). The STE [5] is used to compute the gradient in the backward pass, where the derivative of $q_b$ is ignored and we use the derivatives w.r.t $w_b$ in the SGD optimization to update $w$ iteratively.

The quantization scheme in Eq. (18) involves the clipping parameters $\alpha_w$ and $\alpha_x$, also called scales, that are used to translate the true value of each integer in the network to its floating-point value. This way, at inference time, two integers from different layers can be multiplied in a meaningful way that is faithful to the original floating-point values [31, 26]. These scales also control the quantization error, and need to be chosen according to the values propagating through the network and the bit allocation $b$, whether the weights or the activations.

Alternatively, the works [30, 16], which we follow here, introduced an effective gradient-based optimization to find the clipping values $\alpha_x, \alpha_w$ for each layer. Given Eq. (18) the gradients w.r.t $\alpha_w$, and $\alpha_x$ can be approximated using the STE [30, 16]. For the activation maps, for example, this resolves to:

$$
\frac{\partial x_b}{\partial \alpha_x} =
\begin{cases}
0 & \text{if } x \leq 0 \\
1 & \text{if } x \geq \alpha_x \\
\frac{x_b}{\alpha_x} - \frac{x}{\alpha_x} & \text{if } 0 < x < \alpha_x.
\end{cases}
\tag{19}
$$

This enables the quantized network to be trained end-to-end manner with backpropagation. To further improve the optimization, [30] normalize the weights before quantization:

$$
\hat{w} = \frac{w - \mu}{\sigma + \epsilon}.
\tag{20}
$$

Here, $\mu$ and $\sigma$ are the mean and standard deviation of the weight tensor, respectively, and $\epsilon = 10^{-6}$.

---

[1]We assume that the ReLU activation function is used in between any convolution operator, resulting in non-negative activation maps, and can be quantized using an unsigned scheme. If a different activation function is used that is not non-negative, like $\tanh()$, signed quantization should be used instead.

# D   Stable channel and resolution changing layers

The classical CNN architecture typically applies several layers like Eq. (7). However, to obtain better representational capabilities for the network, the number of channels typically increases every few layers. In CNNs, this is typically accompanied by a down-sampling operation performed by a pooling layer or a strided convolution. In such cases, the residual equation Eq. (7) cannot be used, since the update $F(\mathbf{x}_j, \boldsymbol{\theta}_j)$ does not match $\mathbf{x}_j$ in terms of dimensions. For this reason, ResNets typically include 3-4 steps like

$$\mathbf{x}_{j+1} = \mathbf{K}_j\mathbf{x}_j + F(\mathbf{x}_j, \boldsymbol{\theta}_j), \quad \text{or} \quad \mathbf{x}_{j+1} = F(\mathbf{x}_j, \boldsymbol{\theta}_j), \tag{21}$$

throughout the network to match the changing resolution and number of channels. These layers are harder to control than Eq. (7) and (11) if one wishes to ensure stability.

In this work we are interested in testing and demonstrating the theoretical property described in Sec. 2 empirically, using fully stable networks (except for the very first and last layers). To this end, when increasing the channel space from $n_{c_{in}}$ to $n_{c_{out}}$ for $\mathbf{x}_{j+1}$ and $\mathbf{x}_j$ respectively, we simply concatenate the output of the step with the necessary channels from the current iteration to maintain the same dimensions. That is, we apply the following:

$$\mathbf{x}_{j+1} = \begin{bmatrix} \mathbf{x}_j + hF(\mathbf{x}_j, \boldsymbol{\theta}_j) \\ (\mathbf{x}_j)_{1:n_{c_{out}} - n_{c_{in}}} \end{bmatrix}, \tag{22}$$

where $(\mathbf{x}_j)_{1:n_{c_{out}} - n_{c_{in}}}$ are the first $n_{c_{out}} - n_{c_{in}}$ channels of $\mathbf{x}_j$. This assumes that each of the channel changing steps satisfies $n_{c_{in}} \leq n_{c_{out}} \leq 2n_{c_{in}}$, which is quite common in CNNs. This way, we only use the symmetric dynamics as in Eq. (11) throughout the network, which is guaranteed to be stable for a proper choice of parameters. Finally, we do not apply strides as part of Eq. (22), and to reduce the channel resolution we simply apply average pooling following Eq. (22), which is a stable, parameter-less, operation.

# E   A stable variant of MobileNetV2

In addition to standard residual networks, we also consider the popular MobileNet family of light-weight CNN architectures [36, 25], which are the most common architectures for edge devices, achieving very nice results while requiring modest computational resources to deploy. These architectures utilize the "inverse bottleneck" structure, where the channel space throughout the network is relatively small, but in every step it is expanded and reduced by $1 \times 1$ convolutions, with "depthwise" $3 \times 3$ convolutions in the middle expanded channel space. The depthwise convolutions apply $3 \times 3$ kernels on each channel without mixing between the channels, and hence they are less expensive than $1 \times 1$ convolutions. The general structure of MobileNetV2 [36] reads:

$$\mathbf{x}^{(k+1)} = \mathbf{x}^{(k)} + \mathbf{K}_{1\times1}^3 \sigma(\mathbf{K}_{\mathsf{dw}}^2 \sigma(\mathbf{K}_{1\times1}^1 \mathbf{x}^{(k)})) \tag{23}$$

where $\mathbf{K}_{1\times1}^3, \mathbf{K}_{1\times1}^1$ are two different learnable $1 \times 1$ convolution filters, and $\mathbf{K}_{\mathsf{dw}}^2$ is a learnable depthwise $3 \times 3$ convolution matrix. Due to the relatively small channel space used throughout the architecture, the quantization of this network to low bit-rates is challenging.

The MobileNetV2 architecture in Eq. (23) can also be seen as a residual network, and is generally unstable. Because of its inverse bottleneck structure, it does not fit the structure of Eq. (7) and Eq. (11). To test the importance of stability under quantization for such networks as well, we define the stable MobileNetV2 variant:

$$\mathbf{x}^{(k+1)} = \mathbf{x}^{(k)} - (\mathbf{K}_{1\times1}^1)^\top ((\mathbf{K}_{\mathsf{dw}}^2)^\top \sigma(\mathbf{K}_{\mathsf{dw}}^2 \mathbf{K}_{1\times1}^1 \mathbf{x}^{(k)})), \tag{24}$$

where now, the separable convolution operator $\mathbf{K}_{\mathsf{dw}}^2 \mathbf{K}_{1\times1}^1$ takes the role of the single operator $\mathbf{K}$ in Eq. (11). Going between Eq. (23) and (24) we lost one non-linearity, but as we show later, this has marginal influence on the accuracy of the network. We also apply the depthwise operation twice, but this is inexpensive compared to $1 \times 1$ convolutions. Changing the resolutions and channel space is done the same way as described in section D.

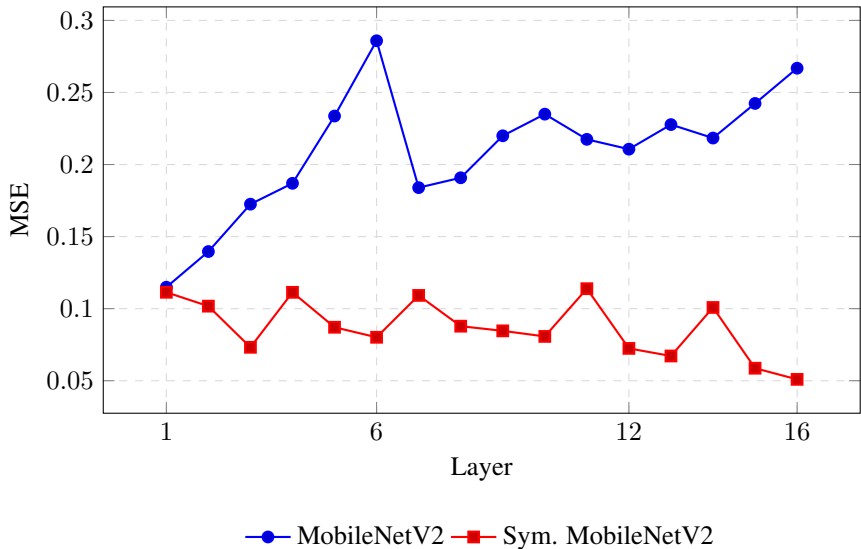

Figure 3: Per-layer MSE between the activation maps of symmetric and non-symmetric MobileNetV2 network pairs - in each pair one network has quantized activation maps and the other does not. Values are normalized per-layer to account for the different dimensions of each layer. In all the cases, the symmetric variants (in red) exhibit a bounded divergence, while the non-symmetric networks diverge as the information propagates through the layers (in blue), and hence they are unstable. Both networks in each pair achieve similar classification accuracy.

## F  Settings and data-sets

Our code is implemented in PyTorch, and all experiments are conducted on an Nvidia RTX-3090 with 24GB of memory. Below, we elaborate on the different data-sets explored throughout the numerical experiments.

**CIFAR-10/100.** The CIFAR-10/100 image classification data-sets [29] each consist of 60k natural images of size $32 \times 32$ where each image is assigned to one of ten categories (for CIFAR-10) or one hundred categories (for CIFAR-100). The data-set includes 50K training examples and 10K test examples. We derive a validation set for training by holding out 10% of the training data and report accuracy metrics on the test data.

**Node-classification data-sets.** Lastly, we use graph neural networks on three citation network node classification data-sets: Cora, CiteSeer and PubMed [37]. For each data-set, we use the standard train/validation/test split as in [38], with 20 nodes per class for training, 500 validation nodes and 1,000 testing nodes. We use the same training scheme as in [9].

