# OpenReview forum: "Quantized convolutional neural networks through the lens of partial differential equations"
_NeurIPS.cc/2021/Workshop/DLDE — DLDE Workshop -- NeurIPS 2021 Poster_

### Official Review · Reviewer_Xgsy · 2021-09-30
**Interesting paper linking the stability of PDEs to neural network quantization**

**Confidence:** 4

**Review:**

Using PDE / ODE stability properties to design CNN / GNN that peform better under feature quantization is a really neat idea.l This is a nice contribution to the field with high technical merit and practical utility.

Several improvements could however be made to the sections dealing with graph neural networks
- GCN generally refers to the model of Semi-Supervised Classification with Graph Convolutional Networks, Kipf and Welling, ICLR17, with GNN being the more common general term for graph (convolutional) neural networks. I assume that this work applies to the more general class of message passing GNNs and not just to the GCN model of Kipf and Welling, otherwise the scope of this work is somewhat limited.
- Missing citations related to diffusion PDEs and GNNs
-- Continuous Graph Neural Networks, ICML20
-- GRAND: Graph Neural Diffusion, ICML21.


**Score:**

4: Very good paper

---

### Official Review · Reviewer_qKoS · 2021-10-11
**Review for Quantized convolutional neural networks through the lens of partial differential equations**

**Confidence:** 2

**Review:**

### Summary

The authors study the stability properties of CNN and GCN models with quantization, the process of reducing computational requirements by reducing the precision of the parameters in a neural network. In particular, error propagation in the network is analyzed through the "lens" of partial differential equations, and the stability properties are considered analogously to the Courant-Friedrichs-Lewy conditions for PDEs. From this analysis, the authors design stable variants of quantized CNNs and symmetric variants of quantized GNNs. Experiments show that these new architectures can yield improvements in accuracy. Moreover, stable and symmetric variants exhibit a per-layer bounded divergence in error propagation compared to their non-stable/asymmetric counterparts.

### Significance of the work

Designing quantized neural networks with stability properties "through the lens of partial differential equations" is an interesting and neat idea. Moreover, this work can bring relevant practical contributions by introducing Pareto optimal neural networks in terms of computational requirements and accuracy for applications with resource or time constraints.

### Other

The paper is well written and structured. Some miscellaneous notes:

- Line 70: the Jacobian $\mathbf{J}$ is not explicitly defined in the main text
- Table 3 (caption): typo "and and"

**Score:**

4: Very good paper

---

### Official Review · Reviewer_6KLc · 2021-10-12
**Nice use of the analogy between discretized DEs and neural architectures to study weight quantization**

**Confidence:** 4

**Review:**

The authors consider the practice of weight quantization in deep convolutional neural networks and deep convolutional graph neural networks. Leveraging the analogy between convolutional architectures and time-dependent PDEs, the authors argue for certain conditions that should improve the stability of network outputs with respect to weight quantization. In actuality, they propose two practices: a specific form of weight-sharing to create a symmetry in the action of convolutional layers, and the use of CFL-like conditions to guarantee the numerical stability of these symmetric layers. This is an ingenious use of the analogy between neural architectures and DEs to understand an increasingly important problem in deep learning (i.e., the need for enormous amounts of computer memory). I recommend the submission be accepted to the workshop.

Discussion:

Line 3: The authors state that fixed-point arithmetic is not natural to neural computations. I am curious to better understand what is meant here. Are there other applications in which fixed-point arithmetic can be incorporated more naturally? Similarly, are there really any applications in which reducing the floating point accuracy of variables can be done without careful precautions? I am no expert in this matter, but it seems to me that deep networks are remarkable robust to weight quantization (compared to, e.g., how I would expect a finite element solver to respond to a reduction from 32 bit variables to 4 bit variables). This has little impact on the rest of the submission, but it piqued my curiousity.

Lines 21-30: This discussion is a little clumsy, given how central this analogy is to the rest of the work. I think the authors' main point is that the forward evaluation of a convolutional network (with residual connections) through its layers is analogous to the forward evolution of a PDE through time. The other points made in the paragraph distract from this message. Similarly, a discussion of the CFL conditions would probably make this more accessible to those members of the DLDE/SciML community who may be less familiar with classical numerical analysis.

Lines 35-47 are good; Lines 48-53 appear to repeat something very similar.

Fig 1: These might benefit from being plotting with a logarithmic MSE axis, since the ratio of MSE between the stable and unstable variants is of primary interest. Also, can the authors explain what happens at the jumps around layers 9 and 16, and why the blue and red curves respond very differently in the intervening layers?

The authors may be interested in the recently released Principles of Deep Learning Theory (https://arxiv.org/abs/2106.10165). That work provides an interesting perspective of the stability of very deep neural networks which may be complementary to the results discussed here.

The submission might benefit from additional discussion of related works, especially regarding the type of symmetry they impose in Equations 2 and 4. To what extent has such symmetry been studied before (e.g., in refs 1,2,21)? In particular, arguably the most compelling result of the submission is that these symmetric layers perform nearly as well as the more general non-symmetric layers, despite consuming half the memory (regardless of weight quantization). The authors conjecture that this advantage is due to smoother optimization (Line 117), although I don't find this explanation convincing. Do the authors have any evidence of this? I was inclined to interpret this behaviour as being due to an inductive bias, similar to the translation-invariance or locality properties of convolutions. Can the authors comment on what inductive biases the symmetric layers may created for the resulting neural architectures?

The authors' experimental protocol could use some clarifiation. What exactly do the columns (4W/8A, etc.) in Tables 1-3 mean? Was quantization effected before, during, or after training? Admittedly, it seems that many of my questions are resolved in the (fairly long) supplemental materials.

Table 3: The authors claim that the symmetric networks preserve their accuracy better under quantization than the nonsymmetric networks. However, it seems to me that the data do not clearly support this argument. For the CiteSeer case, the symmetric vs nonsymmetric accuracies differ, from left to right, by 1.7%, 1.7%, and 1.1%. If anything, the symmetric network has performed slightly worse under quantization that the nonsymmetric one, unless I am missing something. Perhaps the difference in accuracy values is not the best way of comparing the response of these networks to quantization, but if that is the case then the authors should provide a more explicit form of evidence to support the claim that the symmetric/stable networks respond 'better' to quantization (in some suitably defined sense).

Line 115: I believe this is incorrect. The symmetric architectures will generally be less expressive than the nonsymmetric ones. This is similar to the fact that convolutional networks are less expressive than fully general MLPs. As alluded to above, this loss in expressivity is acceptable (and, indeed, beneficial for certain applications) because it corresponds to an inductive bias that restricts the possible functions the network can represent.


**Score:**

4: Very good paper

---

### Decision · Program_Chairs · 2021-10-16

**Decision:**

Accept (Poster)

**Comment:**

This paper draws an analogy between quantisation error in quantised neural networks and truncation error in the numerical solution of differential equations.

All reviewers agree that this is an excellent paper. I agree; the analogy is a good one with the potential for much further exploration.

In addition to the (very substantial) review comments, I would make a few of my own:

For the quantisation error to decay, the authors mandate that the spectral radius of the Jacobian be strictly less than 1. This means that the corresponding dynamical system converges towards a fixed point. In the infinite-layer limit, all possible inputs would produce the same output. (And in any finite approximation there is merely some bias towards this behaviour.) This seems very unlikely to be desirable behaviour, and is (in my opinion) the greatest weakness of the paper.

The authors define stability simply via a Lipschitz condition (equation (1)). Theoretically speaking, at least, this is always true provided the activation function is Lipschitz. It is not clear that this motivation really adds anything.

Nitpick: when separating names (Line 29) it is correct to use an "en dash" (–, obtainable as "--" in LaTeX) rather than a hypen (-).